# Diversity of Pathological Conditions Affecting Pituitary Stalk

**DOI:** 10.3390/jcm10081692

**Published:** 2021-04-14

**Authors:** Łukasz Kluczyński, Aleksandra Gilis-Januszewska, Magdalena Godlewska, Małgorzata Wójcik, Agata Zygmunt-Górska, Jerzy Starzyk, Alicja Hubalewska-Dydejczyk

**Affiliations:** 1Department of Endocrinology, Faculty of Medicine, Medical College, Jagiellonian University, 31-008 Krakow, Poland; lkluczynski@su.krakow.pl (Ł.K.); magdalena.godlewska@student.uj.edu.pl (M.G.); alahub@cm-uj.krakow.pl (A.H.-D.); 2Department of Paediatric and Adolescent Endocrinology, Faculty of Medicine, Medical College, Jagiellonian University, 31-008 Krakow, Poland; wojcik.gosia@gmail.com (M.W.); azygmunt@post.pl (A.Z.-G.); jerzy.starzyk@uj.edu.pl (J.S.)

**Keywords:** pituitary, stalk, hypophysitis, infundibulum, tumour

## Abstract

Pituitary stalk lesions (PSL) are a very rare pathology. The majority of conditions affecting the infundibulum do not present with clinically apparent symptoms, what makes the diagnosis difficult. The recognition might be also complicated by the non-specific and transient characteristics of hormonal insufficiencies. In our study, we retrospectively analysed demographic, biochemical, and clinical long-term data of 60 consecutive, unselected adult patients (34 women and 26 men) with PSL diagnosed in the Department of Endocrinology, Jagiellonian University in Krakow. The diagnosis of PSL were categorized as confirmed, probable, or undetermined in 26, 26 and 8 patients, accordingly. Given the possible aetiology congenital, inflammatory, and neoplastic stalk lesions were diagnosed in 17, 15 and 20 patients, accordingly. In eight cases the underlying pathology remained undetermined. The most common pituitary abnormality was gonadal insufficiency diagnosed in 50.8% of cases. Diabetes insipidus was detected in 23.3% of cases. In 5% of patients the pituitary function recovered partially over time. Stalk lesions were extensively discussed in the context of the current literature. Based on the published data and our own experience a diagnostic algorithm has been proposed to help physicians with the management of patients with this challenging condition.

## 1. Introduction

The pituitary stalk, also known as the infundibulum, connects the hypothalamic median eminence with the pituitary gland. Comprising both blood vessels and nerves, it constitutes an important element in the proper functioning of hormonal regulation. A large variety of pathological conditions can affect the infundibulum. Congenital disorders, in addition to neoplastic and inflammatory/infectious aetiologies, should be taken into consideration [1]. The clinical course of patients with identical pituitary stalk lesions (PSL) varies from asymptomatic disease to rapidly enlarging tumour with accompanying mass effects and loss of pituitary function [2]. The stalk may itself be involved or in addition to other structures by processes originating in the hypothalamus or pituitary gland. The exact incidence of PSL is difficult to estimate; however, the widespread availability of magnetic resonance imaging (MRI), which is the modality of choice for visualizing sellar and parasellar structures, has increased the detection rate of these pathologies. Due to the specific localisation of the infundibulum and potential risks of biopsy, establishing an accurate diagnosis of the underlying disease frequently remains challenging and requires a systematic approach.

## 2. Materials and Methods

In our study, we retrospectively analysed demographic, biochemical, and clinical data of 60 consecutive, unselected adult patients (34 women and 26 men) with PSL diagnosed, treated and followed up between January 2002 and February 2020 in the Department of Endocrinology, Jagiellonian University in Krakow. Patients with childhood onset of the disease were transferred to our Clinic from the Department of Paediatric and Adolescent Endocrinology, Paediatric Institute. The majority of cases were assessed most recently during hospitalisations in 2019 and 2020. In our study, we present data from the time of PSL diagnosis in addition to data from long-term observation of these patients. Terms such as “stalk tumour”, “infundibulum tumour”, “stalk thickening”, “infundibulum thickening”, and “atrophic/hypoplastic stalk” in various forms were used in the search for patients in the computer system supporting Department of Endocrinology.

We categorized the diagnosis of PSL in these patients as confirmed, probable, or undetermined, following methods previously described by Hamilton and Doknic [3,4]. The first group includes patients with disease confirmed via histopathological examination and those with congenital disorders such as pituitary stalk interruption syndrome (PSIS) and septo-optic dysplasia (SOD). A probable diagnosis was based on consistent clinical, biochemical, and imaging findings typical for a given condition or systemic diseases having a pituitary manifestation and positive findings in a tissue sample taken from another region of the body. Patients without a specific pattern of symptoms or laboratory and imaging findings were categorised as having an undetermined diagnosis. Causes of PSL were also assigned to congenital, inflammatory, and neoplastic categories.

The study is a part of a project “Assessment of the health condition of patients with hypopituitarism in Poland” sponsored by Ministry of Science and Higher Education (grant number N41/DBS/000408), approved by the Bioethics Committee of the Jagiellonian University (opinion number 1072.6120.110.219 of 24 April 2019).

### 2.1. Imaging

Stalk pathologies were visualized on MRI scans that were assessed by the radiologists experienced in pituitary pathology working in University Hospital. Pituitary stalk abnormalities in MRI were defined as congenital disorder if the infundibulum was: untraceable, discontinuous or thin (<1 mm). The enlargement of the stalk was defined as a diameters bigger than 2–3 mm compared to the normal stalk measuring between 1.91 ± 0.4 mm at the connection with the pituitary and 3.25 ± 0.56 mm at the optic chiasm on 1.5 T MRI [3,4]. On 3 T MRI obtained scans these values are 2.16 ± 0.37 and 3.35 ± 0.44 mm, respectively [5]. Each pathology was described on the basis of its distinguishing features. Ectopic posterior pituitary is characterized as a spot of increased T1 signal localized at the median eminence or along the stalk with absence of posterior pituitary bright spot [6]. Radiographic features of PSIS include thin or absent stalk, hypoplastic anterior pituitary lobe and ectopic posterior pituitary [7]. The classic image of SOD consists of agenesis of septum pellucidum, hypoplastic optic nerves/optic chiasm as well as pathologies of corpus callosum and pituitary gland (hypoplastic stalk) [8]. Langerhans histiocytosis (LCH) affecting parasellar structures does not present specific features in MRI, but usually loss of the hyperintense signal of posterior pituitary lobe as well as stalk thickening are observed [9]. Enlargement of pituitary gland and hypothalamic mass are rarer demonstrations [10]. Typical radiological manifestations of lymphocytic hypophysitis are enlargement of non-deviated infundibulum with homogenous enhancement after contrast administration, absent bright spot of posterior pituitary and dural tail sign [11]. The variety of MRI features of neoplastic tumours is very large and precise radiological description of these changes is beyond scope of this study. In any case, a systematic approach, taking into account detailed medical history, clinical symptoms and biochemical test results, provides guidance that may assist in the radiological evaluation.

### 2.2. Biochemical Assessment

Hormonal function of anterior and posterior pituitary lobes was assessed in connection with clinical context. Complete data was available for 46 patients. Low levels of peripheral hormones measured from morning samples of blood in correspondence with a lack of adequate increase in trophic hormones were considered as pituitary axes deficiencies. Tests with cosyntropin, insulin, or glucagon were performed to confirm secondary hypocortisolism. In patients with childhood onset of the disease, after qualification based on auxological (growth retardation, growth deficiency, delay in bone age) or biochemical (low level of age and sex adjusted IGF-1 or IGF-1 and IGFBP3) criteria, at least two dynamic tests (insulin tolerance test, glucagon, clonidine or arginine test—in various combinations) were performed during primary assessment in the Paediatric Institute to define growth hormone deficiency (GHD). In the remaining patients, IGF-1 alone was used to define GHD. Diagnosis of diabetes insipidus (DI) was based on basic biochemical measurements and the water deprivation test, if required. If available, hormonal assessment refers to data recorded at the time of diagnosis. Patients with clinical suspicion of systemic disease were tested for abnormal values of IgG4, HCG, AFP, and other neoplastic markers. Additionally, the QuantiFERON test was performed in some cases to exclude tuberculosis.

### 2.3. Statistics

Data were collected and analysed in Microsoft Excel (Microsoft Office 365, Microsoft Corporation, Redmond, WA, USA). Results are shown as numbers with mean value, percentage, or median.

## 3. Results

Sixty patients (34 women and 26 men) were enrolled in our study. The mean age of patients at the time of diagnosis was 33.8 ± 23.7 years (range 1–77 years), 40.9 ± 23.1 years for women and 24.6 ± 21.5 years for men. Confirmed diagnosis was assigned to 14 patients with PSIS (four women, 10 men; mean age at diagnosis was 7.6 ± 5.1 years) and two patients with SOD (one woman, one man; mean age at diagnosis 7.5 ± 7.8 years). Other cases categorised as confirmed diagnosis were patients operated due to pituitary stalk tumours: six pituitary adenomas (two corticotropinomas, two somatotropinomas, one drug-resistant prolactinoma, and one non-functioning tumour with infundibular manifestation), one pituitary carcinoma with extrapituitary metastasis, one craniopharyngioma, one germinoma, and one lesion that was post-surgically verified as lymphocytic hypophysitis. A probable diagnosis was established for 10 patients with clinical symptoms and imaging findings of lymphocytic hypophysitis (six women and four men; mean age at diagnosis was 42.5 ± 20.4 years). Another four patients with Langerhans histiocytosis (two women and two men; mean age at diagnosis was 16.3 ± 15.4 years) and three older women with disseminated neoplasms and metastasis to the pituitary stalk were given a probable diagnosis. Finally, three cases of craniopharyngiomas, one woman with posterior pituitary lobe ectopy, one woman with prolactinoma, one man with granular cell tumour, two patients with infundibular adenoma, and one woman with suspicion of pituicytoma (in the last two, suspicion was based only on MRI) were categorised as probable diagnoses. In eight patients (13.3%), background of the observed stalk pathology remained undetermined.

Given the possible aetiology seventeen patients were categorized as having a congenital cause (14 with PSIS, two with SOD, and one woman with ectopic neurohypophysis) were found, which accounted for 28.3% of all PSL cases. Lymphocytic hypophysitis and LCH (together were 15 cases) comprised the group of inflammatory causes and constituted 25.0% of patients. The neoplastic category was the most heterogenous, as it included nine adenomas, three metastatic tumours, four craniopharyngiomas, one pituitary carcinoma, one germinoma, one pituicytoma, and one granular cell tumour, making up 33.3% (*n* = 20) of PSL. The above data is shown in Table 1.

Clinical symptoms associated with deficits in anterior pituitary hormones led to the initiation of diagnostic work-up in 29 patients (48.3%; including 15 patients (25.0%) with growth retardation). Symptoms related to mass effect (headaches, visual disturbances, and seizures) were seen in 13 patients (21.7%). Polydipsia and polyuria were the initial manifestation in 11 cases (18.3%), while five cases (8.3%) had a clinical presentation of excessive hormone production. Incidental diagnosis was seen in two female patients (3.3%).

Hormonal assessment findings in tested cohort are summarized and presented in Table 2.

### 3.1. Congenital Abnormalities

In the group with congenital abnormalities, mean duration of follow-up was 22.3 years, with the first observation beginning in 1975. The average age of patients at the time of diagnosis was 7.6 years. A broad spectrum of hormonal disorders was described, with GHD confirmed in all cases. Sixteen patients (94.1%) underwent subsequent growth hormone therapy. Hypogonadotropic hypogonadism and central hypothyroidism were the second most common abnormalities (*n* = 14/17, 82.3%, both), followed by secondary adrenal gland insufficiency (10/17, 58.8%). Hyperprolactinemia was detected in 5/17 cases (29.4%). Functioning of the posterior pituitary lobe remained almost always intact, with DI being detected in only one male patient with PSIS in the first year of life (5.9%). Multiple hormonal deficits were observed in 16/17 (94.1%) patients (Table 3). Average age of patients at the time of detection was 7.4 years for thyroid axis dysfunction, 9.3 years for GHD, 13.0 years for central hypocortisolism, and 13.9 years for gonadal axis deficiency. Among cases of initially diagnosed multiple hormonal deficits, GHD was seen in 11 patients (64.7%), while central hypothyroidism was observed in nine patients (52.9%). Growth retardation and symptoms of hypoglycaemia were the most frequent clinical presentations. In 13 cases (76.5%), new pituitary axis deficiencies appeared within a few years after the disease was diagnosed. Figure 1 presents typical pituitary image of patients with PSIS.

### 3.2. Inflammatory Processes

Among four patients (two women and two men) with LCH (Figure 2) mean duration of follow-up was 13.8 years, with the longest observation period lasting 21 years. All four patients initially presented with symptoms of DI. The average age of patients at the time of diagnosis was 15.8 years. Additional hormonal deficits were seen in three patients (75%). Somatotropic and gonadal axis deficiencies were detected in two cases. Secondary adrenal gland insufficiency was observed in one female in this group. One patient was diagnosed with DI as the only pituitary abnormality. Secondary thyroid gland insufficiency and hyperprolactinemia were not found in this group. Hormonal replacement along with specific treatment for histiocytosis remain as the cornerstones of therapy in these patients.

Eleven patients (seven women and four men) presented with suspicion of pituitary stalk thickening due to lymphocytic hypophysitis. Mean duration of observation in this group was 5.2 years, with the first cases having been diagnosed in 2008. The average age at the time of diagnosis was 44.1 years. In three patients, no hormonal deficits were found. In these cases, diagnosis was established based on imaging findings and presenting symptoms alone (headaches in a 71-year-old man and a 27-year-old woman as well as signs of hyperprolactinemia in a 32-year-old woman). The remaining eight patients presented with at least one hormonal deficiency, with DI (4/11—45.5% cases) being the most common. Secondary adrenal gland insufficiency and central hypothyroidism were diagnosed both in 3/11 (27.3%), whereas hypogonadotropic hypogonadism in 2/11 cases. IGF-1 was measured in 8/11 from this subgroup and was found low in two cases. Two patients were treated with steroids (one orally, one intravenously) due to significant neurological symptoms. One patient with inconclusive imaging results of the pituitary stalk and a lack of improvement in subsequent MRI studies underwent surgical operation with histopathological confirmation of lymphocytic hypophysitis. In the remaining group of patients (8/11, 72.7%), a watchful waiting approach was introduced with improvement/stabilisation of pituitary stalk thickening in follow-up imaging (Figure 3). Besides 1 case of probable recurrence, no hormonal deterioration was observed. Table 4 summarizes clinical, biochemical and imaging findings in patients with inflammatory cause of PSL.

### 3.3. Neoplastic Change

The group comprising infundibular neoplastic changes was the most diverse in terms of aetiology. We documented 20 cases (14 women and six men) with a mean follow-up duration of 9.9 years. The longest observation period was 21 years. Mean age of patients at the time of diagnosis was 45.2 years. Almost half of the detected lesions were benign adenomas, with eight of them (two prolactinomas, one corticotropinoma, two somatotropinomas, and three non-functioning tumours) initially presenting with significant involvement of the pituitary infundibulum seen in MRI. In one patient, the recurrence of Cushing disease was located in the pituitary stalk. Furthermore, we identified four craniopharyngiomas, one germinoma, one pituicytoma, and one granular cell tumour. Additionally, there were 3 patients with metastases to the stalk from extracranial neoplasms (Figure 4) and one case of pituitary carcinoma with extrapituitary metastasis. Gonadal axis deficiency was the most common hormonal abnormality in this cohort (11/19 checked patients—57.9%), followed by secondary thyroid (9/20—45.0%) and adrenal gland (5/20—25.0%) insufficiencies. Diabetes insipidus was diagnosed in 25.0% of patients. Assessment of the somatotropic axis was available in 12 cases, of which GHD was detected in three patients (25.0%). Multiple hormonal deficits were found in 11 patients (55.0%). In eight patients higher prolactin values were discovered (Table 5). All hormonal pathologies had a permanent character. From this group, 10 patients underwent surgery, while the remaining patients are under observation or have died due to progression of the primary disseminated neoplastic process. The woman with pituitary carcinoma and metastasis to the other brain and spinal cord structures was treated surgically and then underwent radio- and chemotherapy. Unfortunately, the patient died less than two years after being diagnosed.

Clinical, biochemical and imaging findings in patients with undetermined aetiology of pituitary stalk lesion were shown in Table 6.

### 3.4. Imaging

Imaging studies revealed a thickened pituitary stalk in all patients from the inflammatory and neoplastic subgroups, in one patient with ectopy of the posterior pituitary lobe, as well as in eight patients with undetermined aetiology of PSL (total *n* = 44). A precise initial measurement was available for 35 of these cases, with a mean value of 6.1 mm (range 2.3–15.0 mm). Mean thickness of the pituitary stalk in inflammatory, neoplastic, and undetermined causes were 5.1, 6.3, and 7.7 mm, respectively. Follow-up imaging studies were accessible in 9/11 patients with hypophysitis. The follow-up MRI revealed reduction/stabilisation of infundibulum size in eight cases. One woman underwent surgery. Among patients with histiocytosis, we noted a reduction in stalk size in one patient as well as enlargement in stalk size in another patient. In the neoplastic subgroup, 10 patients underwent surgical therapy after initial assessment due to progression of pituitary pathology in subsequent imaging. Stable stalk lesion size was seen in five patients upon further follow-up, while three women with metastasis to the infundibulum died shortly after diagnosis. Congenital changes characterised by aplastic or hypoplastic stalk were seen in all cases except for one, which was the case of a woman with suspicion of ectopic neurohypophysis.

## 4. Discussion

Pituitary stalk lesions are very rare, there are only few studies in the literature concerning this pathology. In the largest series of patients including 152 cases evaluated in the Mayo Clinic, neoplastic, inflammatory, and congenital changes were found in 49 (53%), 30 (33%), and 13 (14%) cases, respectively. The cause of PSL in 60 patients was not determined [13]. Catford et al. summarised a small group of case series and case reports finding that an inflammatory or infective background was diagnosed in 68% of cases (*n* = 142), while a neoplastic and congenital aetiology accounted for 29% (*n* = 60) and 2% (*n* = 5) of cases, respectively [14]. A study published in Serbia determined that the most common aetiology of PSL was congenital, which was seen in 47.1% of cases (*n* = 25), followed by inflammatory and neoplastic processes, responsible for 16.9% of cases (*n* = 9) each [3]. In our study, neoplastic lesions were detected in 20 cases (33.3%), while congenital anomalies were attributed to 17 patients (28.3%). Lymphocytic hypophysitis and Langerhans cell histiocytosis formed the group of inflammatory causes, which accounted for 25.0% (*n* = 15) of patients. The aetiology was undetermined in 8 cases (13.3%). Our results are in line with data previously presented by Hamilton et al. who described inflammatory/infective, congenital, and neoplastic origins of PSL in 35% (*n* = 23), 34% (*n* = 22), and 31% (*n* = 20, with included histiocytosis), respectively [4]. The relatively large percentage of congenital pathologies in our cohort may be associated with the close cooperation between Paediatric and Adult Endocrinology Departments. It is worth noting that American studies describe sarcoidosis as being among the most significant causes of PSL; however, this is not reflected in European or Asian large series studies [3,4,13,14].

Due to the unique role of the infundibulum, lesions here may cause pituitary dysfunction. In the Mayo Clinic study, 43 out of 152 enrolled patients presented with DI, with at least 1 of the anterior pituitary axes being insufficient in 49 cases (32%), and with hypogonadotropic hypogonadism as the most common deficit [13]. Cai et al. confirmed anterior pituitary lobe dysfunction in almost half of the patients [15]. Furthermore, a high percentage of GHD (84.9%) and central hypogonadism (64.1%), followed by secondary hypothyroidism and hypocorticism (both at 45.2%) were described by Doknic et al. [3]. In our cohort, 45 patients (75.0%, based on the full evaluation of 46/60 patients) had at least 1 hormonal deficit. Gonadal axis insufficiency was the most common abnormality (50.8%), followed by central hypothyroidism (45.0%), GHD (40.0%), and adrenal axis deficiency (31.7%). Somatotropic axis dysfunction may be underestimated due to incomplete data. Hyperprolactinemia and diabetes insipidus were diagnosed in 18 (30.5%) and 14 cases (23.3%), respectively.

The dominant diagnosis in patients with congenital disorders was PSIS, characterised by typical hypothalamic-pituitary structural abnormalities associated with various degrees of hypopituitarism, seen in 14 patients. There is no data regarding the exact prevalence of PSIS. However, Xu et al. observed PSIS in 7.8% (45/577) of cases with confirmed growth hormone deficiency (GHD) [16], while other studies have reported the condition in 4–8% of patients with hypopituitarism [17,18]. A male predominance is consistent with previous observations [19]. The heterogenous clinical presentation of this syndrome may raise diagnostic doubts, and because of this, establishment of a diagnosis is often delayed [20]. Differentiation with post-traumatic damage to the pituitary infundibulum is essential [21]. The remaining cases in the subgroup were SOD and ectopy of the pituitary posterior lobe. All of our patients presented with GHD, while central hypogonadism and hypothyroidism were other frequent abnormalities (82.3%, both). Similar findings were published by Doknic et al. [3]. Dysfunction of the posterior pituitary is a rare manifestation, with DI having been confirmed in only one patient. Adequate hormonal substitution should be introduced as pituitary deficiencies develop. Due to the potentially progressive character of the disease, a thorough follow-up is required in this group of patients.

LCH is a rare disorder of poorly understood aetiology defined by uncontrolled proliferation of cells from the mononuclear phagocyte system [22]. Its annual incidence is estimated at two to five patients per million [23]. LCH often presents multiorgan manifestation and a particular predilection for the hypothalamic-pituitary area Diabetes insipidus is the most frequent endocrine manifestation of the disease, occurring in 12% of patients with LCH. It usually develops within one year, while in 6% of cases it is already present at the time of diagnosis [9,24]. Diabetes insipidus may be also a harbinger of systemic disease [25]. The prevalence of DI reaches 94% in cases with other pituitary hormone deficits. Up to 20% of LCH patients show anterior pituitary involvement, with GHD being the most common abnormality, followed by gonadotropin deficit. Secondary adrenal and thyroid gland insufficiencies are rarer [26]. Our data is consistent with that found in the literature [26]. All patients enrolled in our study were diagnosed with DI and in all cases, it was the first endocrine manifestation. Growth hormone and gonadotropin deficiency were each confirmed in 50% of cases, while secondary hypocortisolism was detected in one patient. No pituitary-thyroid axis pathologies were found. Endocrine abnormalities in LCH patients are almost always permanent, regardless of the type of therapy [27], which was also noted in our long-term follow-up observations.

Primary hypophysitis has an annual incidence estimated at one case per nine million, but the increasing number of cases of IgG4-related pituitary inflammation should alter our perception of the epidemiology of this disorder [28,29]. Lymphocytic hypophysitis is the most common histological type, accounting for approximately 68% of cases [30]. This variant strongly correlated with pregnancy and the pubertal period, with a female predominance (3:1); however, a recent German analysis confirmed the relationship between hypophysitis and pregnancy in only 11% of cases [31]. In our study, we did not document any associations between pregnancy and hypophysitis. Moreover, the female to male ratio was 1.75:1. As with histiocytosis, DI may be the first manifestation of the disease [12] and is present in approximately 39% of patients at the time of diagnosis. Secondary adrenal gland insufficiency is the most frequent anterior pituitary hormonal deficit (60%), followed by gonadotropic deficiency (55%), central hypothyroidism (52%), and GHD (38%) [30]. A study by Honegger et al. reported DI in 54% of cases, with hypogonadotropic hypogonadism as the most frequent pituitary dysfunction (62%) [31]. In our study 45.5% of patients presented DI. Function of thyroid, adrenal, and gonadal axes were disrupted in 27.3%, 27.3%, and 18.2%, respectively. The observed differences might be explained by the relatively small groups of cases. Additionally, insufficient knowledge regarding this disorder among medical specialists could lead to underdiagnosis of the disease. Hormonal substitution is the basis of therapy.

Neoplastic changes in the hypophysis include primary and secondary lesions. Adenomas are the most common among pituitary tumours with a prevalence estimated at 14.4% seen in autopsies and 22.5% as seen in imaging studies [32]. Germinomas are extragonadal germ cell lesions, accounting for 3% of paediatric brain tumours. Typical manifestations include symptoms resulting from mass effect or hypopituitarism, with precocious puberty as a possible manifestation seen in children [33]. Craniopharyngiomas are rare (2–5% of all primary intracranial neoplasms), relatively benign, slowly growing intracranial tumours originating in pituitary gland embryonic tissue. Clinical symptoms are related to the pressure effect on brain structures [34]. Less common lesions such as pituicytomas, tanycytomas, and astrocytomas have also been reported [13,35,36]. Metastatic tumours account for 1% of masses in the sellar region [37]. A study by Javanbakht et al. analysed 289 patients with pituitary metastasis reported in the literature between 1957 and 2018. Breast (*n* = 71), followed by lung (*n* = 68) and thyroid (*n* = 30), were the most common origins of the primary tumours [38]. Little is known about the accurate prevalence of metastasis to the pituitary stalk, but infundibulum along with posterior pituitary lobe are commonly involved [39], what is explained by direct blood supply from inferior hypophyseal artery [40]. Diabetes insipidus (DI) (in 42% of cases) and anterior pituitary hormonal deficits (in 27% of patients) are typical manifestations [41].

In our study, tumours originating from the pituitary gland and involving the stalk were seen in 17 patients (28.3%), while metastases were found in three cases (5.0%). A reliable comparison of data between various studies is challenging due to the significant differences among the described pathologies. An additional challenge is the fact that histiocytosis was included in the group of neoplastic lesions in earlier studies [4]. The percentage of neoplastic changes among all causes of PSL in our study is in line with data from the summary of small case series and case reports published by Catford et al. and to results presented by Hamilton et al. [4,14]. In contrast, a Serbian study reported a neoplastic aetiology of PSL in only 16.9% of cases [3]. In our cohort, a large proportion (45.0%) of adenomas is noteworthy, since in other studies, their proportion is much smaller. In addition, we detected one germinoma. Germ cell tumours with stalk involvement have been reported more frequently in Asian populations [42,43], which is consistent with the historical epidemiology of these disorders [33]. However, new data from studies contradict the geographical variation in prevalence of the disease [44]. Considering secondary neoplastic lesions, it is important to mention two detected cases of gastrointestinal neuroendocrine tumours (NETs) with probable metastases to the pituitary gland. The liver is the main metastatic site in the majority of NETs. Furthermore, involvement of the hypophysis is extremely rare [45,46], but the involvement of the pituitary stalk appears to be unique. This phenomenon might be explained by the significant number of patients with NETs diagnosed and treated in our Department.

A study by Turcu et al. investigated the relationship between MRI characteristics of PSL and their aetiology. A uniformly thickened stalk was present in five out of seven patients with confirmed neurosarcoidosis, while a round infundibulum was described in 11 out of 13 patients with congenital pituitary abnormalities. A stalk lesion involving the pituitary and hypothalamus, as seen on imaging, was the only predictor of hypopituitarism [13]. Despite the limited value of these reports, serial MRIs and repeated clinical assessment remain the basic tools in the management of PSL.

The strength of our study is the high number of consecutive, unselected patients hospitalised in one clinical centre and the long-term observation in majority of cases. The shortcoming is that due to retrospective character of the study and variability in clinical management not all data including hormonal assessment were available.

Based on the published data and our own experience we propose a diagnostic algorithm to help physicians with the management of patients with this challenging condition (Figure 5). 

## 5. Conclusions

A large variety of diseases are associated with pituitary stalk lesions; however, the majority of these do not present with clinically apparent symptoms. Obtaining an exact diagnosis is frequently impossible due to the potential risks of pituitary biopsy. Magnetic resonance imaging remains the most important tool in the detection and management of PSL. Complete hormonal assessment is crucial in patients with infundibular lesions, as hypopituitarism is common. The rarity of this disease makes further multi-centre research essential to increase our understanding of PSL.

## Figures and Tables

**Figure 1 jcm-10-01692-f001:**
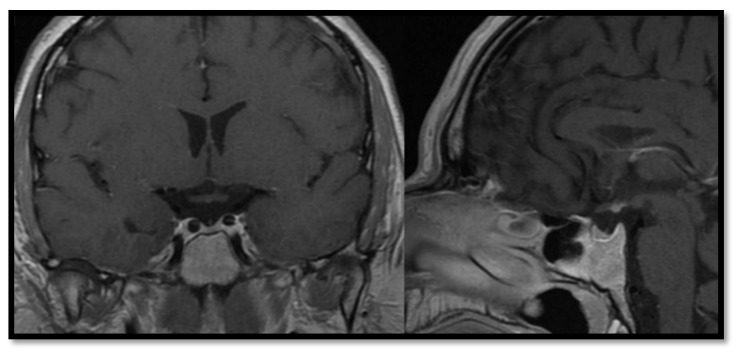
Atrophic infundibulum in pituitary stalk interruption syndrome.

**Figure 2 jcm-10-01692-f002:**
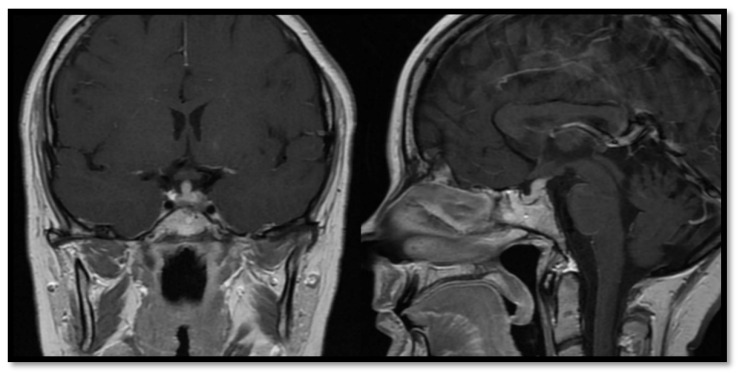
Magnetic resonance imaging of pituitary area in patient with Langerhans histiocytosis.

**Figure 3 jcm-10-01692-f003:**
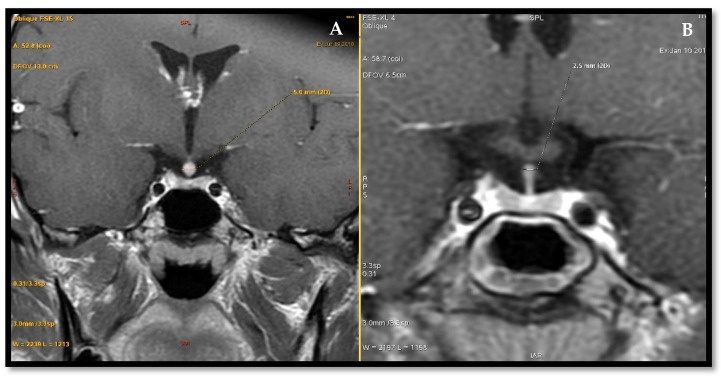
Radiological improvement in patient with primary hypophysitis. Pituitary magnetic resonance: (**A**) June 2018. (**B**) January 2019 [12].

**Figure 4 jcm-10-01692-f004:**
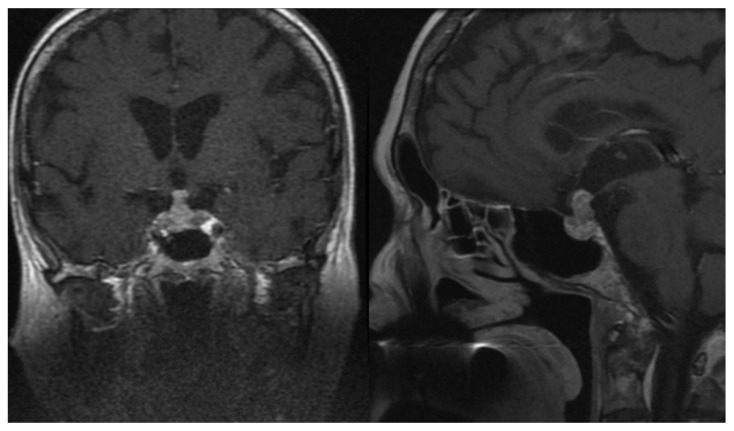
Magnetic resonance imaging in patients with disseminated neuroendocrine tumour and pituitary metastasis.

**Figure 5 jcm-10-01692-f005:**
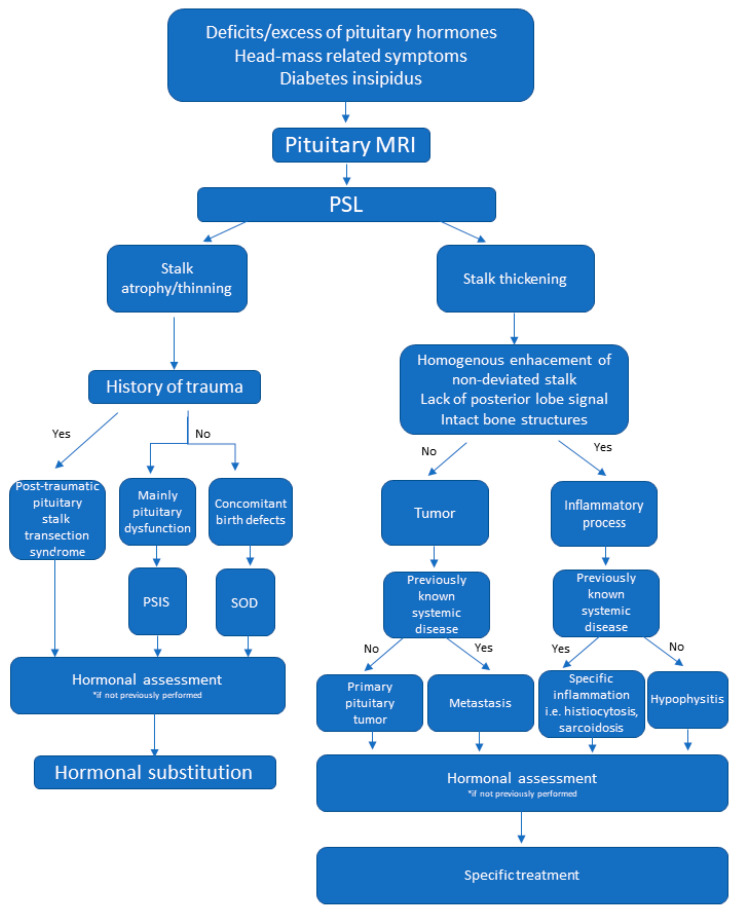
Proposed diagnostic algorithm in patients with pituitary stalk lesions. Abbreviations: MRI—magnetic resonance imaging, PSL—pituitary stalk lesion, PSIS—pituitary stalk interruption syndrome, SOD—septo-optic dysplasia.

**Table 1 jcm-10-01692-t001:** Pituitary stalk lesions aetiology (confirmed/probable in 52 out of 60 of patients; 8 undetermined).

Congenital (*n* = 17)	Inflammatory (*n* = 15)	Neoplastic (*n* = 20)
Pituitary stalk interruption syndrome (14)	Lymphocytic hypophysitis (11)	Adenoma (9)
Septo-optic dysplasia (2)	Langerhans histiocytosis (4)	Craniopharyngioma (4)
Ectopic posterior lobe (1)		Metastatic tumours (3)
		Germinoma (1)
		Pituitary carcinoma (1)
		Pituicytoma (1)
		Granular cell tumour (1)

**Table 2 jcm-10-01692-t002:** Pituitary stalk lesions—hormonal evaluation.

	Adrenal	Thyroid	Gonadal	Somatotropic	Prolactin	Antidiuretic Hormone
Deficiency	19	27	30	24	2	14
Normal/high	41	33	29	23	39	46
					High—18	
No data	0	0	1	13	1	0

**Table 3 jcm-10-01692-t003:** Clinical, biochemical and imaging findings in patients with congenital aetiology of pituitary stalk lesion. (Abbreviations: F—female, M—male, PSIS—pituitary stalk interruption syndrome, SOD—septo-optic dysplasia, MRI—magnetic resonance imaging. Comments: *—during levothyroxine treatment, ^†^—during testosterone therapy; ^$^—during GH treatment).

Number	Patient	Sex	Age of Onset	Diagnosis	Symptoms	Cause	Hormonal Function (1—Preserved, 0—Deficiency, H—High Values)						TSH (uIU/mL)	fT4 (pmol/L)	Cortisol 8:00 (ug/dL)	Max. cortisol in 1 ug Synacthen Stimulation (ug/dL)	ACTH (pg/mL)	PRL 8:00 (uIU/mL)	IGF-1 (ng/mL)	LH (mIU/mL)	FSH (mIU/mL)	Testosterone (nmol/L)	Estradiol (pmol/L)	1st MRI (mm)	2nd MRI (mm)	Comments
							ACTH	GH/IGF1	TSH	LH/FSH	PRL	ADH	0.27–4.20	12.0–22.0	2.3–23.3		15–65	Norms in Brackets	Norms in Brackets	F 2.4–12.6—Folicular Phase, >7.7 Menopause; M 1.7–8.6	F 3.5–12.5—Folicular Phase, >25.8 Menopause; M 1.5–12.4	6.68–25.7 or Norms in Brackets	F 46–607—Folicular Phase; 18.4–201 Menopause			
1	AM	F	14	Confirmed	growth retardation	PSIS	0	0	0	0	1 H	1	0.02	22.31	5.77	x	18.4	3614 (102–496)	28 (191–478)	<0.1	<0.1	x	590.3	atrophic		
2	MC	F	3		growth retardation		0	0	0	0	1 H	1	3.12	10.18	2.64	5.99	40.2	660 (102–496)	31 (191–478)	<0.1	0.27	x	41.04	atrophic		
3	MP	F	4		growth retardation		0	0	0	0	1	1	<0.005	21.15	1.86	x	19	480 (102–496)	33 (191–478)	<0.1	0.13	x	186.7	atrophic		
4	KC	M	3		cryptorchidism		1	0	0	0	1 H	1	1.14	11.82	7.48	20.79	36.3	372 (86–324)	186 (235–408)	4.17	1.42	7.36	x	atrophic		
5	AB	M	9		growth retardation		0	0	0	0	1	1	0.011	14.3	2.12	8.63	581	128 (102–496)	96 (160–318)	0.3	0.3	1.02	x	atrophic		High IGF-1: probably laboratory error
6	MO	M	11		growth retardation		1	0	0	0	1 H	1	1.71 *	14.8 *	9.76	23.5	61.7	472 (86–324)	122 (235–408)	1.23	1.32	16.2 ^†^	x	atrophic		
7	MKn	M	2		growth retardation		1	0	0	1	1 H	1	2.57 *	23.78 *	12.98	21.72	17.1	442 (86–324)	137 (235–408)	4.89	2.73	25.31	x	atrophic		
8	PP	M	11		growth retardation		0	0	0	0	1	1	0.01	19.84	<0.05	x	12.4	97 (86–324)	26 (235–408)	0.23	0.64	14.17 ^†^	x	atrophic		
9	SG	M	6		growth retardation		1	0	1	0	1	1	1.12	18.53	16.92	24.07	73.3	218 (102–496)	65 (154–270)	1.07	1.1	1.08	x	atrophic		
10	ŁC	M	6		growth retardation		0	0	0	0	1	1	<0.005	28.2	4.7	12.93	12	309 (102–496)	219 (154–270)	<0.1	0.35	16.12 ^†^	x	atrophic		High IGF-1: probably laboratory error, patient after GH therapy
11	KK	M	9		growth retardation		1	0	0	1	1	1	1.9	7.72	10.14	x	34.9	80 (30–414)	150 (154–270)	4.1	5.2	3.9 (2.6–10.9)	x	atrophic		
12	SR	M	19		growth retardation		0	0	0	0	0	1	<0.005	25.6	0.2	x	6	69 (102–496)	x	0.9	1.7	9.7	x	atrophic		Lack of IGF-1 result, but patient after GH therapy
13	SD	M	1		hypoglycemia		0	0	0	0	1	0	<0.02	12.5	0.22	x	x	31 (30–414)	223 (235–408)	x	x	5.97	x	atrophic		
14	NS	F	8		growth retardation		0	0	0	0	1	1	2.17	10.6	3.3	10.53	12	81 (102–496)	168 (191–478)	<0.1	0.37	x	<18.35	atrophic		
15	PG	F	2		growth retardation	SOD	0	0	0	0	1	1	0.747	11.5	1.74	x	25.3	457 (102–496)	70 (191–478)	5.21	5.9	x	82	atrophic		
16	EB	M	13		growth retardation		1	0	1	1	1	1	1.17	18.8	14.7	x	24.4	132 (102–496)	417 (235–408) ^$^	3.04	8.17	6.85	x	atrophic		
17	KP	F	16	Probable	growth retardation	Ectopic posterior lobe	1	0	1	0	1	1	2.5	14	11.39	x	x	65 (50–800)	x	2	3	x	<5.0	3	3	Lack of IGF-1 result, patient diagnosed with pituitary dwarfism

**Table 4 jcm-10-01692-t004:** Clinical, biochemical and imaging findings in patients with inflammatory aetiology of pituitary stalk lesion. (Abbreviations: F—female, M -male, DI—diabetes insipidus, MRI—magnetic resonance imaging, OGTT—oral glucose tolerance test, GnRH—gonadotropin-releasing hormone. Comments: * postsurgically, ^†^ proper inhibition of GH in OGTT).

Number	Patient	Sex	Age of Onset	Diagnosis	Symptoms	Cause	Hormonal Function (1—Preserved, 0—Deficiency, H—High Values)						TSH (uIU/mL)	fT4 (pmol/L)	Cortisol 8:00 (ug/dL)	Max. cortisol in 1 ug Synacthen stimulation (ug/dL)	ACTH (pg/mL)	PRL 8:00 (uIU/mL)	IGF-1 (ng/mL)	LH (mIU/mL)	FSH (mIU/mL)	Testosterone (nmol/L)	Estradiol (pmol/L)	1st MRI (mm)	2nd MRI (mm)	3rd MRI (mm)	4th MRI (mm)
							ACTH	GH/IGF1	TSH	LH/FSH	PRL	ADH	0.27–4.20	12.0–22.0	2.3–23.3		15–65	Norms in Brackets	Norms in Brackets	F 2.4–12.6—Follicular Phase, >7.7 Menopause; M 1.7–8.6	F 3.5–12.5—Follicular Phase, >25.8 Menopause; M 1.5–12.4	6.68–25.7	F 46–607—Follicular Phase; 18.4–201 Menopause				
18	IS	F	60	Confirmed	hypopituitarism	Lymphocytic hypophysitis	0	0	0	0	1	1	0.02	11.7	1.66	x	17	140 (102–496)	73 (122–327)	1.4	4.2	x	x	18 × 15 × 12	15	3 *	
19	RK	M	23	Probable	DI		1	0	1	1	1	0	3.67	16.65	15.05	30.66	39.2	384 (102–496)	225 (235–408)	5.34	2.75	9.34	x	4.5	4.5	5 × 5 × 5	2.5 × 3 × 3
20	KP	F	67	Probable	hypopituitarism		0	1	0	0	1	1	0.14	7.47	0.85	x	<5.0	300 (102–496)	105 (91–320)	x	6.1	x	<18.35	3.5	3.5	2.5	2.2
21	BI	M	62	Probable	DI		1	1 H ^†^	1	1	1 H	0	2.19	15.47	8.81	19.09	25.8	422 (86–324)	304 (94–245)	3.84	5.47	9.44	x	4	4	thickened	
22	MG	F	27	Probable	headache		1	1	1	1	1 H	1	2.71	17.88	19.09	x	22.6	598 (102–496)	287 (191–478)	16.4	6.27	x	191	4	4	3	
23	EA	F	51	Probable	DI		1	1	1	1	1	0	0.92	12.76	11.85	x	21.6	417 (102–496)	158 (122–237)	26.94	9.89	x	<18.35	5.5	5.5	3.5	3
24	MK	F	35	Probable	hypopituitarism		1	x	0	1 *	1	1	1.62	11.7	12.57	x	30	149 (102–496)	x	0.52	0.99	x	467.9	3 × 3.5 × 4	3.5		
25	TW	M	71	Probable	headache		1	x	1	1	1	1	3.90	18.58	24.26	x	39.2	84 (50–800)	x	4.98	23.69	24.87	x	thickened	thickened		
26	KB	M	10	Probable	DI		1	x	1	1	1	0	2.51	15	15.3	x	x	252 (102–496)	x	x	x	18.36	x	9 × 6.8 × 4.7	6.8	8 × 3.6 × 3.5	2.5–3
27	JK	F	47	Probable	hypopituitarism		0	1	1	1	1	1	1.10	16.92	0.75	x	14.8	85 (50–800)	178 (123–406)	5.02	5.43	x	230.5	4	4		
28	EK	F	32	Probable	excessive hormone production		1	1	1	1	1	1	2.18	14	17.46	x	27.3	238 (102–496)	401 (180–437)	14.69	3.97	x	465	2.5	2.5	2.5	
29	SJ	M	16	Probable	DI	Langerhans histiocytosis	1	0	1	1	1	0	1.68	15.37	16.25	x	68	226 (102–496)	221 (235–408)	3.54	2.55	26.17	x	4.6 × 4 × 4.6	4.6	3.6 × 2.4 × 3.7	
30	Ekr	F	7	Probable	DI		1	0	1	0	1	0	1.50	12.57	11.96	21.75	19.1	238 (102–496)	109 (191–478)	<0.1	0.46	x	102.6	3	3	4	4.6
31	KA	M	4	Probable	DI		1	1	1	1	1	0	1.94	17.24	21.15	x	22.7	246 (102–496)	309 (235–408)	4.56	4.16	13.44	x	thickened			
32	ES	F	38	Probable	DI		0	x	1	0	1	0	1.53	12.5	6.44	x	17.7	369 (102–496)	x	0.127	1.15	x	46.37	thickened			

**Table 5 jcm-10-01692-t005:** Clinical, biochemical and imaging findings in patients with neoplastic aetiology of pituitary stalk lesion. (Abbreviations: F—female, M—male, MRI—magnetic resonance imaging. Comments: *—whole tumor, ^†^—during therapy with dopamine agonist, ^$^—Klinefelter syndrome).

Number	Patient	Sex	Age of Onset	Diagnosis	Symptoms	Cause	Hormonal Function (1—Preserved, 0—Deficiency, H—High Values)						TSH (uIU/mL)	fT4 (pmol/L)	Cortisol 8:00 (ug/dL)	Max. Cortisol in 1 ug Synacthen Stimulation (ug/dL)	ACTH (pg/mL)	PRL 8:00 (uIU/mL)	IGF-1 (ng/mL)	LH (mIU/mL)	FSH (mIU/mL)	Testosterone (nmol/L)	Estradiol (pmol/L])	1st MRI (mm)	2nd MRI (mm)	3rd MRI (mm)	4th MRI (mm)
							ACTH	GH/IGF1	TSH	LH/FSH	PRL	ADH	0.27–4.20	12.0–22.0	2.3–23.3		15–65	Norms in Brackets	Norms in Brackets	F 2.4–12.6—Folicular Phase, >7.7 Menopause; M 1,7–8,6	F 3.5–12.5—Folicular Phase, >25.8 Menopause; M 1,5–12,4	6.68–25.7 or Norms in Brackets	F 46–607—Folicular Phase; 18.4–201 Menopause				
33	JC	F	44	Confirmed	diabetes insipidus	Pituitary carcinoma	1	x	1	x	1	0	0.98	14.3	36.96	x	31	724 (50–800)	x	x	x	x	x	9	9	10 × 8 × 9	12 × 10 × 11
34	BP	M	30		hypopituitarism	Germinona	0	0	0	0	1 H	1	1.7	7.54	0.96	x	11	696 (102–496)	123 (235–408)	1.01	0.64	<0.09	x	7	7		
35	RS	M	57		headache	Craniopharygioma	1	x	0	0	1	1	0.9	5.9	7.2	26.2	17.4	159 (50–800)	x	0.6	1.8	1.4 (2.6-10.9)	x	11	11		
36	Mka	M	51		hypopituitarism	Prolactinoma	0	1	0	0	1 H	0	0.01	14.64	0.27	x	9.6	83648 (86–324)	212 (144–286)	<0.1	0.62	0.09	x	8	7	11	18 × 14
37	KM	F	71		headache	Corticotropinoma	1	0	0	0	1	1	1.21	12.1	25.38	x	108.7	335 (102–496)	42 (91–320)	0.3	0.3	x	135	7 × 6 × 5	6		
38	AC	F	21		excessive horomne production	Corticotropinoma	1	1	0	0	1	0	0.1	15.27	7.63	18.12	38	119 (102–496)	204 (191–478)	<0.1	<0.1	x	587.3	thickened			
39	MN	F	25		excessive horomne production	Acromegaly	1	x	0	0	1	1	0.07	18	19.2	x	x	229 (102–496)	x	3.6 (4–12)	3.9	0.7	x	8	8		
40	Abo	F	57		excessive horomne production	Acromegaly	1	1 H	1	1	1	1	0.76	12.8	14.86	x	15.7	324 (102–496)	538 (122–237)	20.5	66	x	x	thickened			
41	KZ	F	28		headache	Adenoma	1	1	1	1	1	1	1.48	14.18	15.56	x	57	401 (102–496)	412 (191–478)	2.94	6.46	x	123.5	21 × 14 × 19 *			
42	Mgu	F	57	Probable	hypopituitarism	Prolactinoma	0	x	1	0	1 H	1	0.19	13.4	5.7	x	25	608 (40–470)	x	x	0.59	x	x	thickened			
43	JKo	F	63		confusion	Craniopharyngioma	1	1	0	0	0	1	0.37	9.98	8.38	20.7	18	7 ^†^ (102–496)	214 (91–320)	0.3	0.46	x	38.57	28 × 25 × 18 *			
44	KD	F	22		hypopituitarism	Craniopharyngioma	1	0	1	1	1 H	1	1.09	15.54	23.1	x	22.8	922 (102–496)	142 (191–478)	9.94	5.73	x	119	3 × 5	3	3.5	3.8
45	MS	M	22		seizure	Craniopharyngioma	1	1	1	1 H ^$^	1	1	3.11	17.86	11.4	x	25.8	237 (102–496)	341 (235–408)	19.44	27.91	7.09 (8.64–29)	x	2.3	2.3	2.3	
46	DZ	F	61		incidental	Adenoma	1	1	1	1	1 H	1	2.9	14.79	19.88	x	134	690 (102–496)	116 (91–320)	21.73	35.33	x	x	8 × 5 × 3.6	5	7.5 × 4.6 × 6	6 × 4.5 × 6
47	DP	F	60		incidental	Pituicytoma	1	x	1	1	1	1	2.37	13.4	12.84	24.19	18.1	286 (102–496)	x	11.5 (>10)	53	x	x	11 × 8 × 9	8	11 × 9 × 11	
48	PM	M	28		headache	Adenoma	1	x	1	1	1	1	12.8	13.3	11.4	x	60	117 (102–496)	x	3	4.5	3 (2.6–10.9)	x	7 × 6 × 8	8	7 × 6 × 8	
49	GP	F	68		diabetes insipidus	NET metastasis	1	x	0	0	1 H	0	0.01	12	18.4	x	15.1	887 (102–496)	x	3.3	4	3	x	6	6		
50	MKr	F	74		hypopituitarism	NET metastasis	0	1	0	0	1 H	1	0.04	8.29	0.3	x	16.3	823 (102–496)	125 (91–320)	<0.1	0.25	x	169.4	7.5 × 12 × 6	7.5		
51	JT	F	60		hypopituitarism	Acromegaly +lung cancer metastasis	0	1 H	1	0	1 H	1	0.19	14.1	1.27	12.79	17.3	928 (102–496)	452 (122–327)	<0.1	0.28	x	x	4	4		
52	MH	M	35		diabetes insipidus	Granullar cell tumor	1	x	1	1	1	0	1.57	12.56	20.08	x	27.8	228 (102–496)	x	9.25	7.36	9.44	x	3	3	4	4.1

**Table 6 jcm-10-01692-t006:** Clinical, biochemical and imaging findings in patients with undetermined aetiology of pituitary stalk lesion. (Abbreviations: F—female, M—male, MRI—magnetic resonance imaging. Comments: * whole tumor).

Number	Patient	Sex	Age of Onset	Aetiology and Diagnosis	Symptoms	Hormonal Function (1—Preserved, 0—Deficiency, H—High Values)						TSH (uIU/mL)	fT4 (pmol/L)	Cortisol 8:00 (ug/dL)	Max. cortisol in 1 ug Synacthen Stimulation (ug/dL)	ACTH (pg/mL)	PRL 8:00 (uIU/mL)	IGF-1 (ng/mL)	LH (mIU/mL)	FSH (mIU/mL)	Testosterone (nmol/L)	Estradiol (pmol/L)	1st MRI (mm)	2nd MRI (mm)	3rd MRI (mm)	4th MRI (mm)
						ACTH	GH/IGF1	TSH	LH/FSH	PRL	ADH	0.27–4.20	12.0–22.0	2.3–23.3		15–65	Norms in Brackets	Norms in Brackets	F 2.4–12.6—Folicular Phase, >7.7 Menopause; M 1.7–8.6	F 3.5–12.5—Folicular Phase, >25.8 Menopause; M 1.5–12.4	6.68–25.7 or Norms in Brackets	F 46–607—Folicular Phase; 18.4–201 Meno Pause				
53	MB	F	56	Undetermined	headache	1	1	1	1	1	1	1.3	18.05	36	x	82	167 (102–496)	156 (14–286)	24.49	42.12	x	<18.35	15 × 11 × 10 *	11		
54	EW	M	58		headache	1	x	1	1	1 H	1	0.9	16	19.3	x	x	353 (35–330)	x	7.9	12.5	x	x	5	5		
55	Mgo	F	43		headache	1	x	1	1	x	1	x	14.86	17.54	x	27	x	x	6.92	4.48	x	x	12 × 11 × 7	11	12 × 11 × 7	11 × 10 × 7
56	Dpa	F	77		headache	1	1	1	1	1	1	1.3	16.33	24	x	112	196 (102–496)	96 (91–320)	39.02	82.7	x	x	11 × 7 × 9	9		
57	KK	M	57		headache	1	1	1	1	1	1	4.9	12.59	18	x	21.7	325 (102–496)	134 (94–245)	6.37	5.18	6.3 (2.6–10.9)	x	7 × 5 × 7	7	7 × 5 × 7	
58	Mse	M	25		hypopituitarism	1	1	1	0	1	1	1.9	13.57	41.7	x	29.2	449 (102–496)	339 (235–408)	0.92	0.38	0.77	x	thickened			
59	EM	F	28		excessive hormone production	1	1	1	1	1 H	1	1.49	14.04	27.8	x	17.7	670 (102–496)	270 (191–478)	6.42	6.01	x	341.8	3	3		
60	ZS	F	64		hypopituitarism	1	1	0	1	1 H	1	1.95	8.11	19.04	x	35	827 (102–496)	192 (91–320)	15.55	31.78	x	<18.35	8	8	9 × 10 × 8	

## Data Availability

The data presented in this study are available on request from the corresponding author.

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
