# Peer review of "Diversity of Pathological Conditions Affecting Pituitary Stalk"

_jcm, 2021, doi:10.3390/jcm10081692_

Round 1

Reviewer 1 Report

In this paper, the authors retrospectively describe a cohort of 60 adult patients with pituitary stalk lesions diagnosed in a single center between 2002 and 2020. They give the proportion of the different aetiologies and report biological features and follow-up when available. Their results are in accordance with previous reports. In the present form, this descriptive paper does not give new clues for the diagnosis and treatment of this challenging condition. Major points : - Pituitary stalk lesions are evidenced by MRI. In this series, patients presented with different types of pituitary stalk MRI abnormalities. These MRI abnormalities are the only common feature of the patients selected for this study. It is thus very important to have a clear description of the characteristics of the lesions and to know how imaging may influence the diagnosis work-up and help for the aetiological diagnosis. On this aspect, the informations given in the paper are very scarce. Additionally, it would be more informative to present the characteristics of MRI for each class of aetiology (congenital, inflammatory, tumoral) rather than in a signgle chapter - Imaging has much progressed over the last 20 years. Were the MRI reviewed by an expert radiologist ? - The authors classify the patients as having a congenital, inflammatory or neoplastic aetiology of PSL. The diagnosis is often challenging especially for the differenciation between inflammatory and neoplastic processes . Can the authors precise the criteria used to assess diagnosis ? - Could the authors based on their own experience and the literature propose an algorythm to help physicians with the management of patients in this challenging condition ? Minor points : - The abstract describes the methodology of the study but does not give any result - A lot of abbreviations are used in the text which are not explained (PSIS, SOD) - The technics used for hormonal assessment are not given - IGF1 may be normal in adult patients and dynamic tests are usually required for diagnosis - The results section should be more structured and give more details on the clinical characteristics of patients : initial presentation, mode of discovery of the pathology, evolution etc….A table summarizing the main clinical features, MRI characteristics and evolution may be useful

Reviewer 2 Report

In this paper, Kluczyński et al. reports data concerning Pituitary Stalk Lesions (PSL) in a series of 60 patients over a period of nearly 20 years.

The overall manuscript is interesting for the clinicians which will have to deal with this kind of (rare) lesions, however, data are not that well presented and therefore unclear for the reader.

1)The clinicial, biochemical and imaging features of the population have to be better presented and a table summarizing all of these criteria could help the author to improve its presentation

2) Authors describe the protocol for biochemical screening performed in patients, however there are no data regarding the absolute values. Since the authors present data of deficiency they have to provide the hormonal assessment of each patients

3) I would suggest the authors to present some MRI pictures of PSL to illustrate the manuscript (e.g. before and after a treatment for a neoplasia)

4) 1 prolactinoma was classified in the exact diagnosis beacuase surgically resected. Why this patient has not been treated by dopamin agonist ?

5) There are no data about level of prolactin encountered in the 60 patients, while the one can assume that prolactinemia increases with the pituitary stalk effect. Please clarify this point

Reviewer 3 Report

The present paper presents 60 patients with pituitary stalk lesions, a rare radiological finding. Its main interest is the number of patients, that should allow drawing some interesting conclusions. However, after its reading, some limitations come out to obtain such conclusions, due to the retrospective nature and, most of all, due to variability in clinical management. These limitations should be further discussed in the Discussion section.

I find a major concern in the analysis of the results. As it is stated in the Materials and Methods section, pituitary stalk lesions are categorized in three groups: exact diagnosis (maybe confirmed diagnosis would be a more appropriate term), probable diagnosis and undetermined or unestablished diagnosis (I would try to be consistent and use the same term along the article). But, in the Results section, when presenting the clinical characteristics of the series (clinical presentation, hormonal and radiological evaluation and long-term observation) 9 among the 17 patients in the third group are given a diagnosis (a presumed diagnosis, I would say) and analysed with the patients with a confirmed and a probable diagnosis, as congenital, inflammatory and neoplastic.

Those 9 patients must be analysed together with the 8 other patients with an undetermined diagnosis. Actually the long-term observation of this group is of upmost interest, since sometimes it is difficult for clinicians and patients to maintain a watchful waiting when the diagnosis is unknown, assuming the risks of missing neoplastic diagnosis, growing lesions or new hormonal deficiencies. How many lesions grew? How many regressed? How many new hormonal deficiencies were diagnosed? Comments can be made about those presumed diagnoses and explanations should be given about the clinical and radiological characteristics that rose those suspicions. Table 1 should be redone taking this in mind.

In another vein, being a paper based on a radiological finding, it would gain much more interest if several figures with the MRI images of the different diagnostic categories were included.

Other minor concerns:

Materials and Methods:

  • Could you specify where did you search for the searched terms?
  • Could you specify how many brain MRIs were done in that period? It is not a true population incidence, but it is interesting to know how frequent pituitary stalk lesions are found in clinical practice.
  • The first time an acronym appears in the text (PSIS, SOD, GHD…), it should follow the words it stands for.
  • Could you please specify the IGF1 criterion that was used to define growth hormone deficiency? Was it age and sex adjusted?
  • Page 2,line 85. “Data was collected” should be changed for “Data were collected”

Results:

  • The term “pituitary cancer” needs clarification. Do you mean pituitary carcinoma, with extrapituitary metastasis? No comments on this patient are made in the Long-term observation paragraph. 
  • Page 5, line 159: “The somatotrophic axis was tested in 9/11 patients from this subgroup”. This sentence suggests that stimulation tests were done, which is not true. I suggest “IGF1 was measured in 9/11 from this subgroup and was found low in one case”.

Discussion

  • Page 7, line 272: there are some words missing in the sentence.

Round 2

Reviewer 1 Report

The paper is more clear and informative and is much improved

Reviewer 2 Report

I compliment the authors with this revised version of the manuscript